# Manufacturing and Analysis of Natural Fiber-Reinforced Thermoplastic Tapes Using a Novel Process Assembly

David Hartung [1,*], Stefanie Celevics [1], Patrick Hirsch [1], Ivonne Jahn [1], Lovis Kneisel [2], Kay Kölzig [2], André Matthes [3] and Holger Cebulla [3]

1   Fraunhofer Institute for Microstructure of Materials and Systems, 06120 Halle (Saale), Germany
2   SachsenLeinen GmbH, 04416 Markkleeberg, Germany
3   Institute of Lightweight Structures, Chemnitz University of Technology, 09111 Chemnitz, Germany
*   Correspondence: david.hartung@imws.fraunhofer.de

**Abstract:** The natural fiber-reinforced thermoplastic tape was produced using a novel process assembly that involves a drawframe and a double belt press. First, the state-of-the-art film-stacking process was modified through the integration of a drawframe to supply the natural fiber preforms for reinforcement, adding thermoplastics films as matrix material and processing them to a unidirectional tape (UD tapes) using a double belt press. Based on that, a new approach was investigated using a commingled sliver containing natural reinforcing and polymer matrix fibers to manufacture UD tapes. This leads to a reduced flow path of the matrix polymer, which is a decisive parameter for production efficiency. To ensure a homogeneous distribution and alignment of the fibers after gilling, the influence of various processing parameters on one another and the resulting UD tape quality were examined. As result, a draft ratio in the range of $10 \pm 2$, a low linear density (here 12 ktex) and general use of many thin in contrast to fewer heavier slivers is advisable. The differences in impregnation quality and thus the mechanical performances of the UD tapes from both processes were validated using scanning electron microscopy and mechanical testing. It was found that the commingled sliver composite had 10% higher flexural modulus and 34% higher flexural strength compared to the film-stacking-based composites. In conclusion, using commingled sliver could enable the increase in productivity and fiber volume fraction compared to film-stacking-based composites.

**Keywords:** plant fibers; green composites; sustainable materials; processing methods; properties

## 1. Introduction

Due to the strong increase in environmental pollution worldwide and the consequential climate changes observed [1], the population's awareness of sustainable action is increasing [2]. To do justice to the desire and responsibility of the current generation to leave a habitable planet for future generations, the development of environmentally friendly and sustainable materials is of crucial importance [3]. In this context, the use of conventional fiber-reinforced plastics (FRP) with glass and carbon fiber reinforcements is critical, since energy-intensive processes are necessary for the production of reinforcing fibers and for the processing of FRP, which generates a large $CO_2$ footprint [4]. Accordingly, recyclable and biodegradable material solutions, especially for the use of plastics and their composites, must be established. To substitute synthetic materials with nature-based alternatives, plants that were yet very helpful in human history can be used. Natural fiber-reinforced plastics (NFRP) offer great potential to counter this problem. Vegetable fibers such as flax and hemp absorb around 1.3–1.4 kg of $CO_2$ per kg of biomass from the earth's atmosphere during their natural growth [5,6] and are ideally suited for reinforcement due to their good mechanical properties (due to the high cellulose content) in combination with the low density. Many other natural fibers are considered in various studies, for example, kenaf, coconut, sisal, banana or pineapple, with regional reference and motivation in most cases [7]. The quality

and performance of natural fibers can be influenced in many ways from the extraction of the fiber (soil quality, weather conditions, mechanical pulping, roaster, cleaning), further processing (possible damage) and pretreatments, to the application in the composite material (ambient conditions, load cases) [8,9]. Even incineration, as the last recovery method, promises advantages because natural fibers have approximately the same energy content as lignite [10]. Moreover, natural fibers leave a significantly lower proportion of ash 1–2% compared to glass fiber-reinforced plastics (GFRP) up to 30% after incineration [10]. Since a material made from regenerative raw materials does not guarantee a sustainable solution, standard tools such as the life cycle analysis (LCA) have become established for holistic assessment [11]. Such analysis was used by Le Duigou et al. [12] to compare the production conditions of 1 kg flax and glass fibers about their ecological balance for use in composites. Based on the environmental indicators (climate change, acidification, non-renewable energy consumption, etc.), the results show that hacked flax is an attractive and more sustainable alternative to the use of glass fibers. Referring to the LCA, it is generally assumed that natural fibers are significantly more environmentally friendly compared to glass fibers, due to reduced $CO_2$ emissions and lower energy consumption during the production of the fibers and a subsequent efficient use phase [13,14]. The decisive criteria for the suitability of materials for lightweight construction are the specific mechanical properties of the materials (mechanical properties normalized to density) [15,16]. The low density of 1.4 $g/cm^2$ of bast fibers compared to glass fibers with 2.54 $g/cm^3$ compensates for a lower mechanical performance [16]. In this respect, some NFRP are already approaching the level of GFRP [17,18]. However, some main problems can be identified referring to the assertiveness of NFRP against synthetic material solutions. Conventionally used long natural fibers for continuous reinforcement are first converted into twisted yarns and are usually further processed into fabrics. These textile processing steps are very cost-intensive; thus, these semi-finished products have, to date, not been competitively priced with other material solutions such as GFRP as lightweight material solutions [19,20]. In addition, the helical configuration of twisted yarns lowers the strength and modulus of the fiber itself (imperfect alignment) and affects the resulting composite properties as well because of poor impregnation and consolidation quality due to less porosity between the fibers [21]. To counter this, material and technological solutions must be found. Reliable and cost-effective raw materials supply and an efficient, continuous production line is necessary. To date, the range of applications for natural fiber composites has been largely limited to non-structural elements [21], which are often made from nonwovens. It is still problematic to efficiently produce high-performance and structure-bearing components from natural fiber-reinforced plastics. In addition to the high costs for textile processing of semi-finished products from natural fibers [20], a lack of technological maturity for processing such as decortication and efficient retting [22–24] as well as natural fluctuations of available natural fiber qualities must be taken into account [14,25]. Finally, a well-balanced price–performance ratio is of decisive importance for the large-scale industrial use of NFRP. The price level of technical flax fibers as a side-product from long fiber production at 1.5 EUR/kg used in this work is cheap compared to 7 EUR/kg for the long flax fiber quality [26,27]. Thus, the fiber raw material can be cost-neutral to the E-glass fiber [14,20,25]. The shortage of synthetic raw materials is already leading to noticeable price increases, whereas the price level for technical natural fiber as a regenerative raw material source can be kept stable [28]. Referring to the general use of natural fibers to reinforce plastics, some major challenges, in addition to the costs, such as hydrophilicity, durability, variability in fiber properties, fire resistance and thermal stability can be derived from the literature [14,24,25,29–31]. Suitable and possible applications and the potential of NFRP are described in [6–9,20,24,25,27,29,32–37].

To boost the ecological and economical advantage of natural fibers as reinforcement materials, a novel method of manufacturing high-performance natural fiber-reinforced composites was developed, using technical fibers and an innovative process assembly [14,38].

## 2. Materials and Methods

### 2.1. Material Characteristics

The utilized combing flax fibers were supplied by Madex[®] (Malbork, Poland), with a measured mean length of 128 mm. The unmixed flax sliver went through two drafting operations before being further processed into composite tapes, supplied with a linear density of 14 ktex.

Polypropylene (PP) film (Castolie PPR L200865/C1Mc) with a density of 0.91 g/cm$^3$ used for the process was purchased from POLIFILM GmbH (Weißand-Gölzau, Germany). A melt flow index of 5 g/10 min (190 °C, 2.16 kg) was determined according to ISO 1133. The thickness of the film was 80 μm, with a width of 280 mm and an areal density of 72.5 g/m$^2$, which was the decisive variable for dimensioning the UD tapes.

Textured polypropylene fibers with a length of 80 mm used for the novel processing of UD tapes with commingled sliver were sourced from the Sächsisches Textilforschungsinstitut e.V. (Chemnitz, Germany), with a measured melt flow index (MFI) of 15 g/10 min (190 °C, 2.16 kg) according to ISO 1133. The fibers were provided as slivers with a linear density of 6 ktex. The melting temperature of PP is generally assumed at 160 °C [39]. Table 1 summarizes the properties of the used composite components.

**Table 1.** Properties of the starting materials.

| Properties | Units | Flax Fiber | PP Fiber | PP Film |
|---|---|---|---|---|
| Mean fiber length | mm | 128.00 | 80.00 | - |
| Materials density | g/cm$^3$ | 1.50 | 0.91 | 0.91 |
| Melting temperature | °C | - | 160.00 | 160.00 |
| Melt Flow Index | g/10 min | - | 15.00 | 5.00 |
| Linear density | g/m (ktex) | 12.00, 18.00, 24.00 | 6.00 | - |
| Areal density | g/m$^2$ | - | - | 72.50 |

### 2.2. Methods of Manufacturing

#### 2.2.1. Processing of Fibrous Preforms Using a Drawframe

Unidirectional alignment of the reinforcing fibers in the NFRP leads to optimal substance utilization [40]. A drafting system was used to provide a flat fibrous web out of added slivers (pre-yarn textile product) as a preform to manufacture fiber-reinforced plastics [41]. The porous structurednatural fiber preform can be combined with polymer films, fibers or powder and further processed into a unidirectional composite tape. The used drawframe type gillbox features a double needle bar system made for processing bast fibers and can clean, equalize and align the fibers through implemented rotating combs [42–44]. Commercially available semifinished products from flax are summarized in [36].

#### 2.2.2. Film-Stacking-Draft Process

The manufacturing process, which includes the merging of the material components as well as the impregnation, consolidation and solidification, can be carried out with low effort and high productivity. In Figure 1, the process assembly is shown schematically. For calculation, the slivers linear density (LDS) is transformed into areal density (ADF) after leaving the draft unit. For the actual manufacturing, the number (NB) and linear weight of slivers (total input) were fed to the drawframe in dependency on planned areal density (output) and draft ratio. After the dry preform was encased with films and had entered the double belt press, temperature and pressure were applied to melt the polymer, impregnate the flax fibers and consolidate as well as solidify the composite UD tape that was winded up afterward. A double belt press type *KFK-E* flatbed laminating machine by Maschinenfabrik Herbert Meyer GmbH was used.

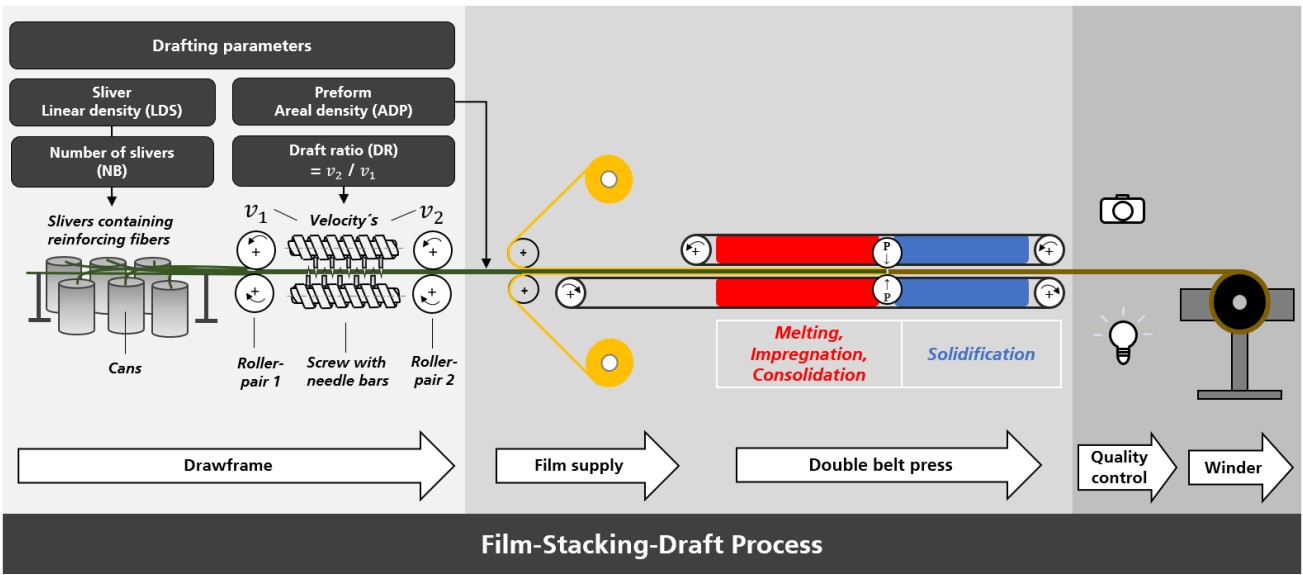

**Figure 1.** Schematic film-stacking-draft process assembly for UD tape production with an additional camera system for optical analysis of the impregnated composite.

The pressure of the double belt press can be regulated indirectly through the nip height of the belts (air cushion system with max. 0.20 bar) and directly through a press roller pair (max. 8.00 bar), placed in the middle of the plant. The process and material parameters are summarized in Table 2.

**Table 2.** Process and material parameters of the film-stacking-draft process.

| Parameters | Values |
|---|---|
| Production speed | 1.00 m/min |
| Heating temperature | 190.00 °C |
| Air cushion pressure | 0.10 bar |
| Roller pressure | 1.00 bar |
| Melt and consolidation time | 81.00 s |
| Solidification time | 69.00 s |
| Fiber/Matrix percent by weight | 50 %/50 % |
| Areal density flax | $145.00 \pm 5$ g/m$^2$ |
| Areal density PP film | $2 \times 72.50$ g/m$^2$ |

### 2.2.3. Commingled-Sliver-Draft Process

The approach of this process methodology is the upstream mixing of finite polymer matrix and natural reinforcing fibers. In contrast to the described film-stacking-draft process, matrix fibers are used instead of matrix films in a commingled-slivers-draft process (CSD). Semi-consolidated UD tapes have been developed in [40,45] with the same approach and formed the basis of this research. The technology of using upstream mixed commingled slivers to produce UD tapes was first developed by Sedlec [45]. In his work, the commingled slivers were heated with infrared radiators to melt the matrix with infrared heaters and subsequently pressed by rollers to spread and impregnate the natural fibers (flax and hemp). Akonda et al. presented a similar process in which a roller draft system was used in combination with infrared heating and a roller consolidation unit to produce UD tapes [40]. In contrast to the technology of Sedlec and Akonda, a gillbox was used for the preform production in combination with a double belt press for the application of heat and pressure in this study. Blending of the reinforcing and matrix fibers to commingled slivers can be

carried out using a conventional carding process, and the optimal alignment and mixing ratio are carried out through drafting. Adapted from [45], the flax and PP fibers were blended into commingled slivers with a ratio of 50:50 weight-%. The processing assembly is depicted in Figure 2.

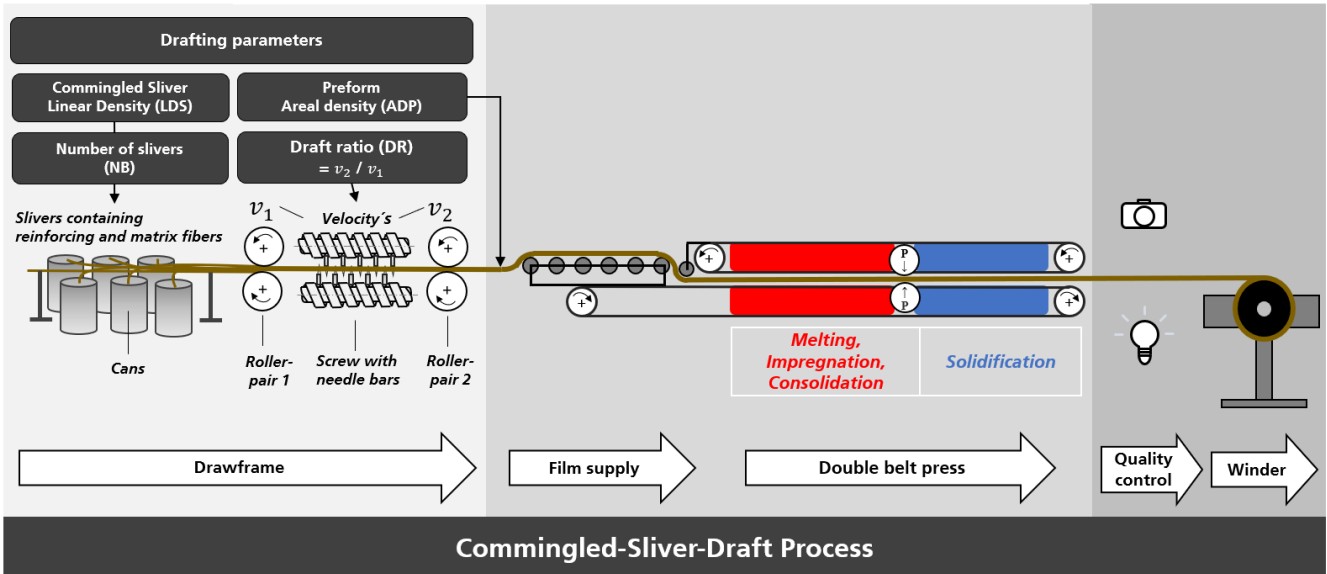

**Figure 2.** Schematic commingled-sliver-draft process assembly for UD composite tape production.

A defined number of commingled slivers were then drafted to provide a preform containing reinforcing and matrix fibers. A key challenge of the process was to guide the unbonded preform, which is sensitive to stress because the fibers are untwisted [45], from the drawframe to the double belt press.

Permanent tension must be applied to guarantee a predominantly unidirectional orientation of the fibrous preform. This tension is necessary to ensure a clean process through static friction, but a distortion of the fiber structure caused by floating fibers has to be avoided (dynamic friction). The tension is limited by the static friction forces of the used fibers [42]. However, no investigations were carried out in this regard. The process parameters are summarized in Table 3.

**Table 3.** Process and material parameters from the commingled-sliver-draft process.

| Parameters | Values |
|---|---|
| Production speed | 1.00 m/min |
| Heating temperature | 190.00 °C |
| Air cushion pressure | 0.10 bar |
| Roller pressure | 1.00 bar |
| Melt and consolidation time | 81.00 s |
| Solidification time | 69.00 s |
| Fiber/matrix percent by weight | 50 %/50 % |

It should be noted, that in addition to the surface roughness, the entanglement and interaction of the fibrils must be considered as well. After entering the press, heat and pressure were applied to the commingled preform. Subsequently, the matrix fibers melt down and impregnate the flax fibers while the pressure consolidates the composite before solidification occurs in the cooling zone to produce a UD tape that was winded up afterward.

### 2.2.4. Lamination of Composite Tapes

For mechanical tests according to standard regulations, laminates with a thickness of 1 and 2 mm were prepared for tensile and bending tests. Therefore, the slim tapes measuring approximately 200 μm thickness were stacked in a 0° direction and laminated using a double belt press with a pressure of 1 bar, temperature of 190.00 °C and velocity of 1.00 m/min. From the fabricated FSD tapes, only FSD.3, FSD.6 and FSD.9 (the ones with the best results in structural analysis) and the CSD tape were processed to laminates.

### 2.3. Evaluation of Drafted Fiber Preforms

A combined analysis of both presented processes could not be carried out due to machinery limitations while processing the commingled slivers. While running the CSD process, reinforcement and matrix fibers must pass through the drawframe. These thick slivers tended to clog the gillbox. To reach the maximum output, the draft ratio could not be varied and was kept to a minimum of 5.2, ensuring stable processing.

### 2.3.1. Test Series Overview and Calculation

The aim and subject of this work is the subsequent processing the preform after leaving the drawframe. Uniformity, faultlessness and permeability of the fibrous web preform is of decisive importance for the later properties of the composite. Since there have been no studies on the structural influence of these fiber architectures on reinforcement qualities to date, an attempt was made to analyze the fiber structure using an optical measurement system. The aim was to achieve a uniform spread of the fiber input slivers to secure as much free fiber contact surface area as possible for optimal impregnation with the matrix component, while simultaneously achieving homogeneous mass distribution as well as straightened fiber orientation. According to Kruger, two to three gilling operations (GO) are necessary to properly align and disentangle the individual fibers. In addition, the first GO is recommended with a high draft ratio for better initial alignment [41]. Thus, the used slivers have been gilled three times in total. Table 4 provides an overview of the film-stacking-draft (FSD) test series carried out with the considered parameters and their characteristics. To ensure comparability, the areal density of the output preform (ADP), with a width of 200 mm, remained constant. Input draft parameter variations summarized in Table 4 were calculated using Equation (1).

$$ADF\,(const.) = \frac{LDS \cdot NB}{DR \cdot 0.20\;m} \tag{1}$$

**Table 4.** Film stacking test series overview. With draft ratio (DR), number of slivers (NB), the linear density of slivers (LDS), gilling operations (GO).

| Tape ID | DR | NB | LDS (ktex) | GO |
|---------|-------|-------|------------|----|
| FSD.1 | 5.80 | 12.00 | 12.00 | 3 |
| FSD.2 | 8.60 | 18.00 | 12.00 | 3 |
| FSD.3 | 11.60 | 24.00 | 12.00 | 3 |
| FSD.4 | 5.80 | 8.00 | 18.00 | 3 |
| FSD.5 | 8.60 | 12.00 | 18.00 | 3 |
| FSD.6 | 13.00 | 18.00 | 18.00 | 3 |
| FSD.7 | 5.80 | 6.00 | 24.00 | 3 |
| FSD.8 | 7.70 | 8.00 | 24.00 | 3 |
| FSD.9 | 11.60 | 12.00 | 24.00 | 3 |

The total UD tape grammage results by adding the areal density of the fibrous flax web (ADP: $145.00 \pm 5$ g/m$^2$) and the film areal density ($2 \times 72.50$ g/m$^2$) together. The calculated output was 290.00 g/m$^2$ with a ratio of approx. 50:50 weight-% flax and PP.

### 2.3.2. Method of Optical Analysis

The following parameters were investigated for their influence on the fibrous flax web quality: input linear density of slivers (LDS), input number of slivers (NB) and the draft ratio (DR). For the evaluation of the processing parameters, quality attributes of the fiber structure have been worked out. Figure 3 shows characteristic defects in the fiber architecture of a dry preform. Elaborated quality criteria for the later comparison are marked in Figure 3. By using gray value distributions, image analysis was a promising and easily applicable method to mark out differences in the fiber structure depending on the draft parameters.

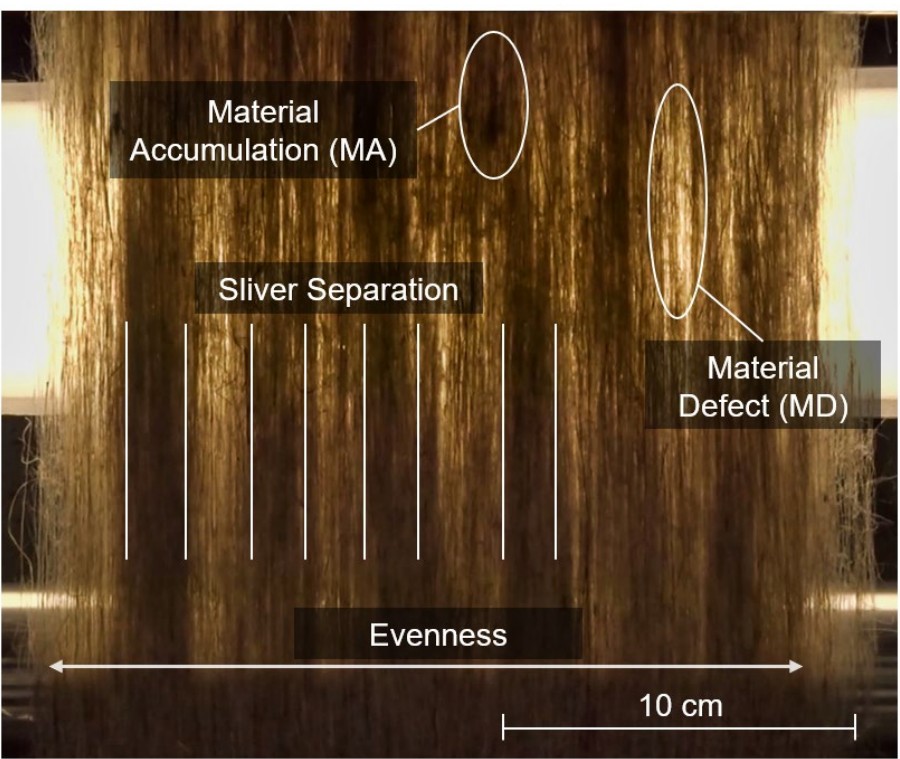

**Figure 3.** Characteristic features of the fibrous web as quality criteria of the impregnated UD tape.

The characteristic features of the fiber structure were operationalized into measurable optical parameters, summarized in Table 5.

**Table 5.** Operationalization of the quality criteria for the assessment of the fiber structure.

| Optical Quality Criteria | Measurable Parameters from Gray Value Distributions |
| --- | --- |
| Evenness of fiber structure | Average mean gray values (AGV) and its coefficient of variation (CV) |
| Occurrence of material accumulations and defects | Area share of certain grayscale areas: material accumulations (MA), material defects (MD) |
| Separation of single slivers | Contiguous grayscale areas |

The assessment of the fiber architecture, which is the focus of this series of tests, was primarily carried out through optical analysis and mechanical characterization. For this purpose, a camera (Canon 70D with a fixed focal length lens of 50 mm) was installed at the

exit of the double belt press to obtain video material for further evaluation. Since shadows had to be avoided, a flat LED panel was used to illuminate the fiber structure. The image settings of the camera remained identical for all tests.

The raw video footage was used to extract images periodically every 30 s with a total of 10 picture samples per test. These were then processed using the open-source software GIMP® (version 2.10.22) created from the GIMP team (USA) The size of the image samples was set to $500 \times 900$ pixels ($15.00 \times 27.00$ cm with around 85 ppi) with a total of 45,000.00 measurement points. For later image analysis with open source software ImageJ® (version 1.53) created from the National Institutes of Health (Bethesda, MD, USA), the samples were decolorized in GIMP® by changing the picture modus to grayscale. In this case, the gray values were calculated using linearized sRGB: Luminance = $(0.22 \times R) + (0.72 \times G) + (0.06 \times B)$. Further information on the calculation method can be looked up in [46].

## 3. Results and Discussion

### 3.1. Preform Quality

3.1.1. Preliminary Parameter Analysis

Since the varied investigation parameters can only be considered independently from one another to a limited extent (with constant areal density output), it is assumed that a mutual influence of the parameters causes the observed effects to interact, i.e., there is multicollinearity between them. The results of the multiple linear regression models are given in Table 6. For the calculation of linear regression models, multicollinearity must be avoided to obtain interpretable results and reliable conclusions. Hence, the variance inflations factors (VIF) were calculated within a multiple linear regression model that contains all explanatory variables (no matter which response variable is related). The result clearly shows multicollinearity with DR = 9.84, LDS = 9.20, and NB = 18.84 (a value over 10 is considered critical). Thus, the variable with the highest coefficient (NB) was taken out of the model. With NB excluded, the VIF coefficients were calculated again to DR = 1.00 and LDS = 1.00, which allows the resuming of interference statistics.

**Table 6.** Multiple linear regressions models.

| | Multiple Linear Regression Results | | | |
|---|---|---|---|---|
| | *Dependent variable:* | | | |
| | z_AGV | z_StdDev | z_MA | z_MD |
| | (1) | (2) | (3) | (4) |
| z_DR | 0.914 *** | 0.467 | −0.847 *** | 0.873 *** |
| z_LDS | 0.241 | 0.674 ** | −0.134 | 0.276 |
| $R^2$ | 0.872 | 0.642 | 0.725 | 0.816 |
| Adjusted $R^2$ | 0.829 | 0.523 | 0.633 | 0.754 |
| F Statistic (df = 2; 6) | 20.450 *** | 5.391 ** | 7.899 ** | 13.275 *** |

Note: * $p < 0.1$; ** $p < 0.05$; *** $p < 0.01$.

To evaluate the effect dominance of draft ratio (DR) and the linear density of slivers (LDS) on the measured dependent variables and to equalize value ranges of the independent variables, a z-standardization was carried out before calculating multiple linear regression models. All models are significant and can be further interpreted. The DR shows a strong influence on average gray value (AGV) and the occurrence of material accumulation (MA) and material defects (MD). LDS rather determines the degree of scattering and slenderness as measured through the standard deviation(StdDev) of AGV.

3.1.2. Optical Analysis

The picture preparation explained in Section 2.3 with grayscale distribution of the FSD tapes was used to analyze and evaluate the quality of the reinforcing fiber structure. Grayscale histograms were used to evaluate the influence of process parameters on the

preform morphology. The gray value range is from 0–255, respectively, from dark to light (256 gray levels as 8-bit depth). For each test series, the raw data of the individual histograms of each image sample were averaged and a new histogram was generated from the mean values of the samples. These averaged histograms are shown in Figure 4. It can be seen that the distribution changes by altering the processing parameters. For a low draft ratio, a right-skewness distribution is measured, which is caused by material accumulations, and thus there is little influence of light. The following measurements were all carried out with ImageJ® for image analysis. The differences in the grayscale distribution between the test series are recognizable. By comparing the individual histograms, various effects can be detected. An evenly distributed set of values appears in the form of a normal distribution, which can be represented as a density function of defined features (in this case, the gray values). The degree of the slenderness of this density function depends on the standard deviation. In the optimal case, each pixel would be assigned the same gray value and thus the same local material density (apart from natural color differences of the fiber material).

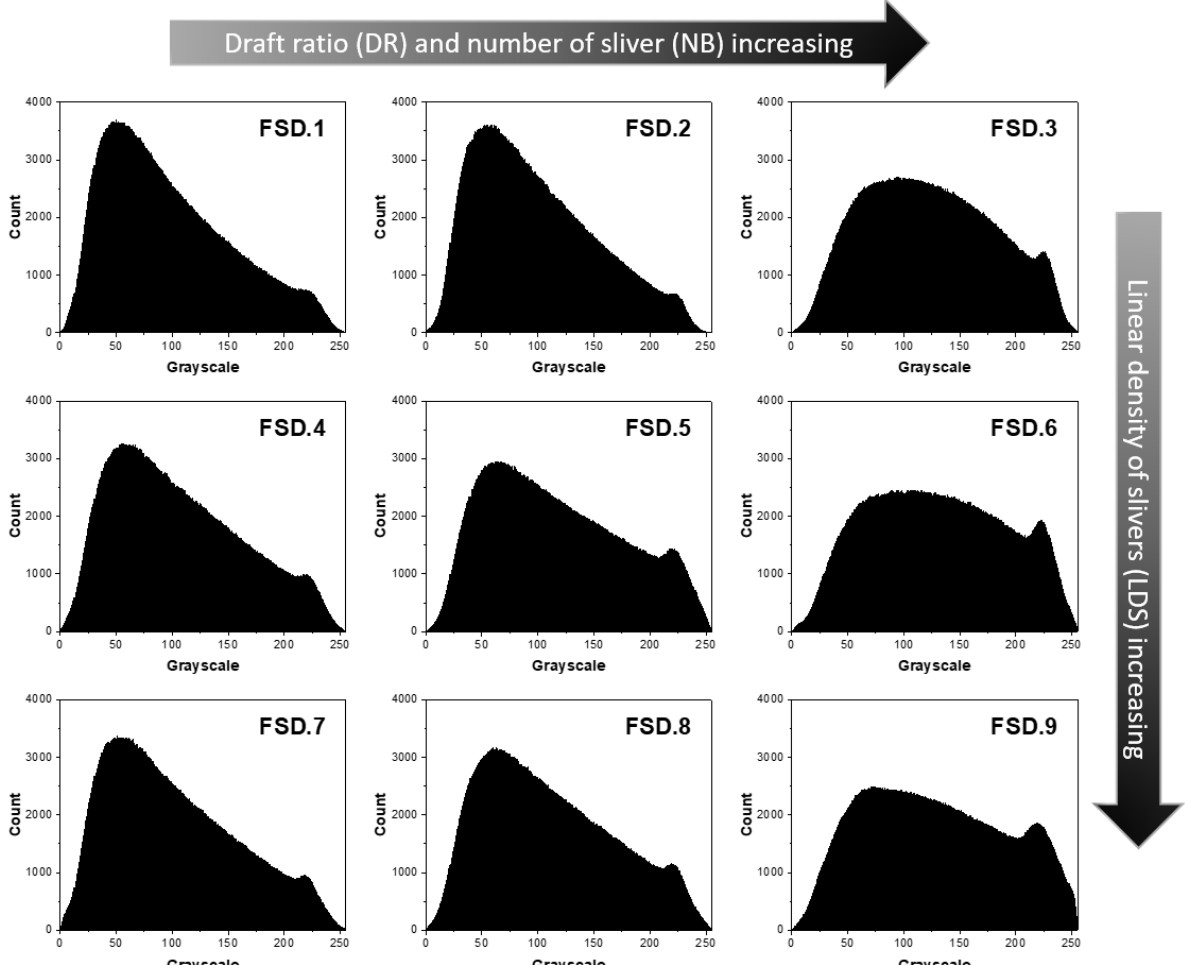

**Figure 4.** Averaged grayscale histograms of film-stacking-draft composites in comparison.

The density function should therefore be as slim as possible and distributed symmetrically around an optimal value. As 127 is the center of the gray value distribution, it is further assumed as the optimum of the gray value distribution. To assess the evenness of the fiber structure, the AGV is shown in Figure 5a. To evaluate the occurrence of MA and MD, the margin areas of the grayscale were measured and compared in Figure 5b. The regions of interest were set to 50 gray levels each.

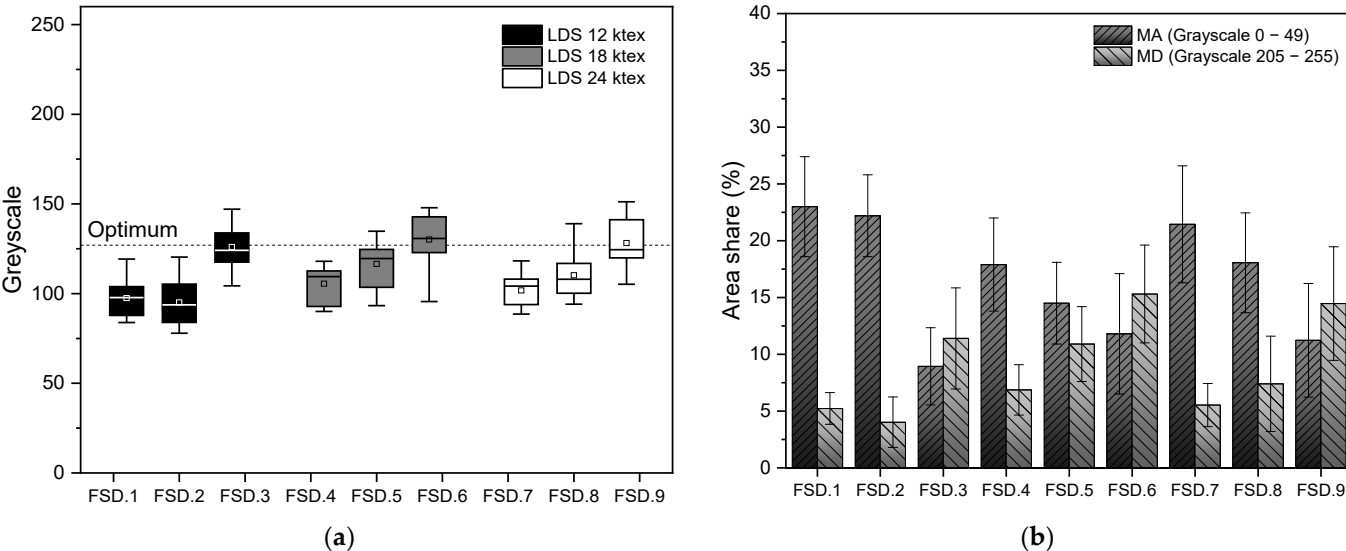

**Figure 5.** Comparison of the FSD test series: (**a**) averaged gray value (AGV) in relation to the assumed optimum; (**b**) results of the averaged area share of material accumulation (MA) and material defects (MD) using threshold function with ImageJ® software. Comparing the histograms in Figure 4 and the distribution of AGV in Figure 5, an analogous behavior can be observed. By increasing DR and NB, AGV shifts towards the optimum 127 in all test groups. The best results are shown in groups FSD.3, FSD.6 and FSD.9. In Figure 6a, the influence of the draft ratio on the average gray value and its coefficient of variation is depicted. Two ambivalent trends occur: increasing DR concludes with a rising AGV (up to the suggested optimum and above) while the CV of AGV is simultaneously decreasing. These effects are both considered to be positive for the quality of the uniformity of the preform and the later UD tape, respectively, because the more fiber surface is accessible, the better the impregnation quality of the composite will be. The accessible surface area of the fibers will increase by shifting the AGV toward the assumed optimum gray value. This happens by eliminating the structure and presence of single slivers due to high draft ratios. In addition, the coefficient of variation as the scattering measure of the averaged gray value gives information about the uniformity of the drafted fiber structure. A low CV describes a homogenous morphology. Regarding the appearance of MA and MD, clear trends can be derived from Figure 6b: as the draft ratio increases, the proportion of material accumulations decreases, and material defects increase. Accordingly, the smallest area proportions of MA occur in the test series FSD.3, FSD.6 and FSD.9 with the highest DR. The comparison of MD follows the ambivalent trend. The occurrence of MD increases by increasing DR. When both effects are considered, the best results were obtained with high DR and low LDS (and consequently fewer NB).

Overall, FSD.3 with the highest number of slivers of 24, a draft ratio of 11.6, and the lowest LDS of 12 ktex shows the best results under the condition of an inevitable compromise between eliminating material accumulations and the prevention of material defects. The draft ratio seems to be the decisive influence on the uniformity of a drafted fibrous web. If the draft ratio is too high, a lot of material defects appear, which is not tolerable. However, a high draft ratio is necessary to effectively eliminate the material accumulations and single-sliver structures. In [44], the authors measured the drafting force with a detecting bar and a load cell. With this, a continuous signal of the bar's position could be recorded which was changing due to sliver irregularities. This made it possible to correlate drafting force and sliver irregularity, as well as drafting force and draft ratio. The results showed that drafting force and sliver irregularities are proportional to each other, they correlate positively. In addition, the reciprocals of the draft ratio and drafting force are proportional to each other and correlate positively. This confirms the assumption, that higher draft ratios lead to fewer irregularities in the output fibrous web. Furthermore, the drafting process result is influenced by the fiber length, roller gauge, acceleration behavior,

falling needle bars or material properties such as the stiffness of the fibers, which have not been considered regarding the analysis [42,47–50].

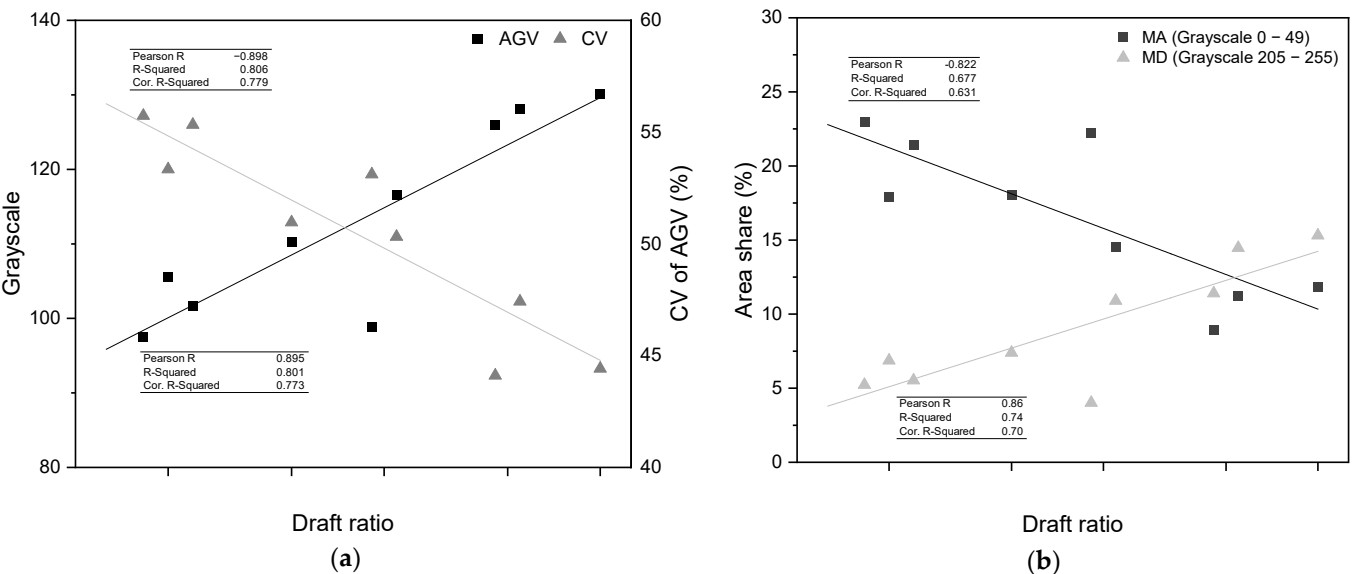

(**a**)　　　　　　　　　　　　　　(**b**)

**Figure 6.** Correlation between draft ratio (DR), average gray value (AGV) and its coefficient of variation (CV) as well as material accumulation (MA) and material defects (MD): (**a**) AGV and its CV in relation to DR, (**b**) averaged area shares of MA and MD of the total area vs. DR.A structural delineation of individual slivers in the fibrous web is clarified by two aspects: due to the high material density, the areas are delimiting dark, and they are arranged next to each other in a contiguous straightway. One option to highlight differences between the investigated UD tapes is provided by the "Wand" function (tracing tool) in ImageJ®. With this tool, the program can recognize connected structures by specifying a tolerance value of the color (or gray) value; thus, the selection can be adjusted accordingly. Figure 7 presents the results of the use of this function with high (left) and poor visibility (right) of the slivers for a set tolerance value of 30. The example on the left is more likely to show structural delineation due to high linear density and low number of slivers as well as a low draft ratio. The higher the tolerance value is, the larger the recognized related image parts are. By increasing the tolerance value, the related image areas become larger. By decreasing, the analysis can be refined. This easy method could be part of a quality management software solution with an automated control mechanism.

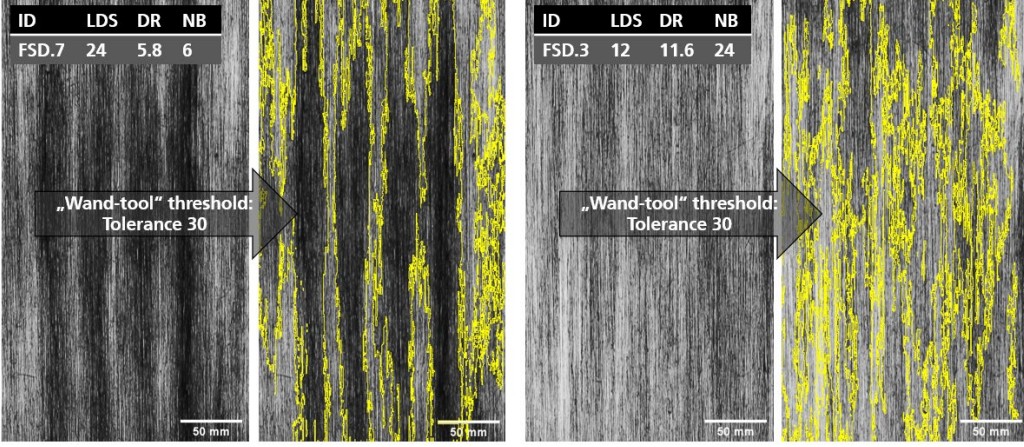

**Figure 7.** Visibility of contiguous gray-level areas as structural delineation of individual slivers. The example on the left clearly shows the contour of individual slivers while the example on the right is more equalized through favorable processing parameters.

### 3.2. Comparison of Film-Stacking-Draft and Commingled-Sliver-Draft Composites

3.2.1. Microscopic Analysis

To examine and compare the morphology of the tapes, a scanning electron microscope (SEM) was used. Two magnifications for each (800× shows the whole tape cross-section (left) and 5000× a region of interest (right)) tapes are shown in Figure 8. As expected, significantly fewer voids can be seen in CSD tape compared to FSD tape since air and steam from natural fiber moisture (no pre-drying was performed) had more possibilities to be eliminated from the CSD process. Fiber outlines with matrix voids probably appear due to the evaporation of embedded moisture and subsequent shrinkage of the fiber during cooling. Imperfect mixing of reinforcing and matrix fibers results in positions with matrix excess as well as missing matrix. The surface of the CSD tape is often poorly impregnated compared to that of film-stacking tape because of missing initial polymer. In contrast to this, the film-stacking tapes show matrix excess at the edges and dry fibers and voids in the middle region of the tape, where fiber aggregation is appearing, and the polymer melt penetration is hampered.

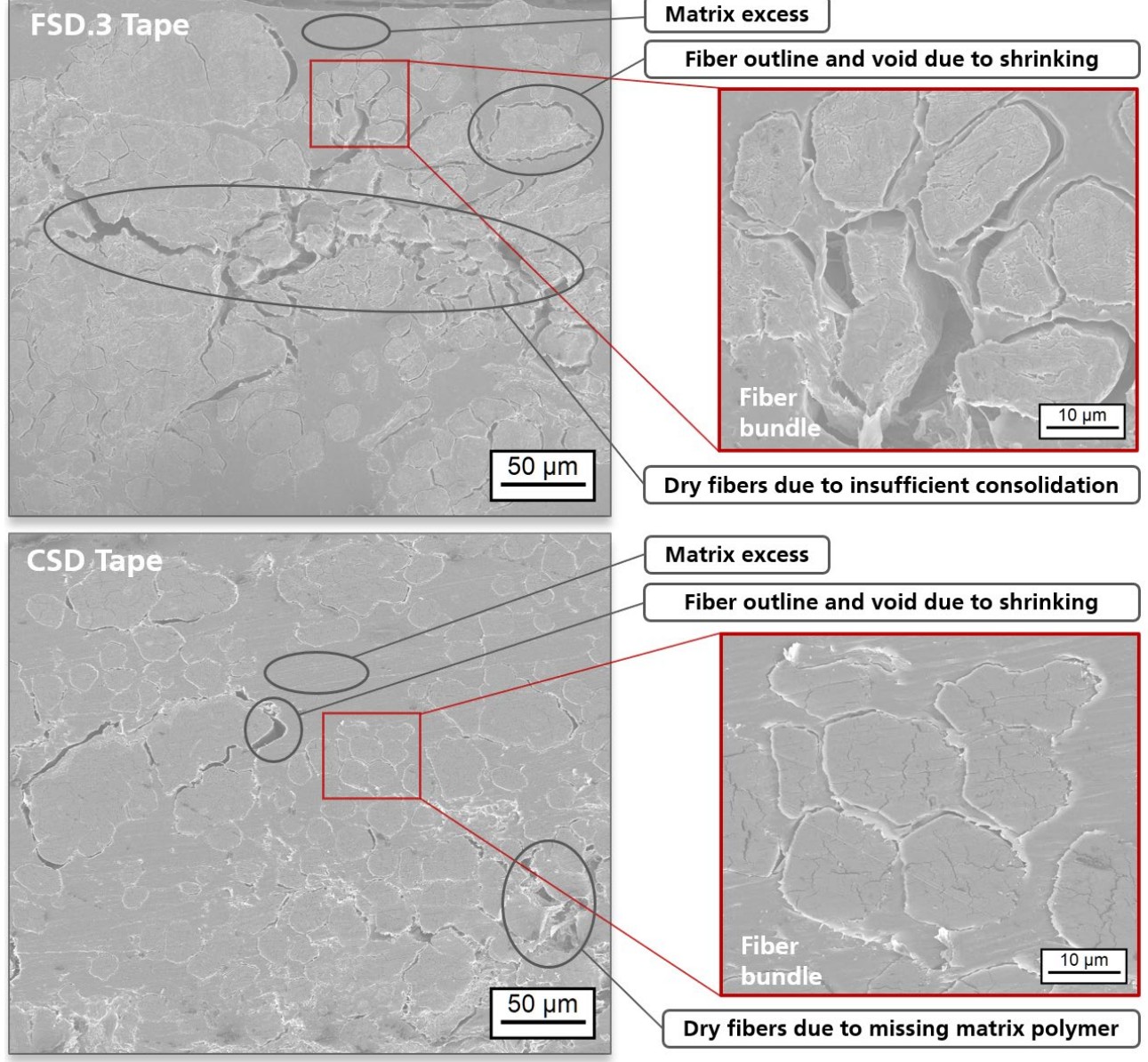

**Figure 8.** Images from scanning electron microscope: FSD.3 tape (**top**), CSD tape (**bottom**).

Thus, the two melt fronts do not meet and mingle. Unlike the film-stacking-draft process, micro impregnation dominates while processing the commingled slivers [51]. The flow path that can be saved for the macro impregnation by using mixed reinforcing and matrix fibers benefits the micro impregnation and elimination of voids, whereas processing with matrix films is dominated by macro impregnation because maximum distance has to be covered from the outside to the center. This relation is depicted schematically in Figure 9. Despite the identical production parameters, a higher impregnation and consolidation degree was achieved using CSD, which results in overall higher tape quality. However, the incomplete closed edge of the CSD tape compared to the FSD tapes (where a lot of matrix material remains on the outside) is problematic. To counter this, a combination of CSD and FSC process (additional films or nonwovens), fiber spraying or powder scattering would be conceivable. General findings that can be derived from the comparison of commingled-sliver-draft and film-stacking-draft processing are that the reduced initial flow path of the matrix system (fibers instead of films) leads to minimum time needed for macro impregnation and subsequently more available time for micro impregnation [51]. This ensures a more complete impregnation which, in turn, increases the mechanical properties. Moreover, it should be noted that the viscosity of polymer fibers is generally lower than for film polymer in terms of processing requirements [52], this factor also lowers the needed time for proper impregnation and consolidation.

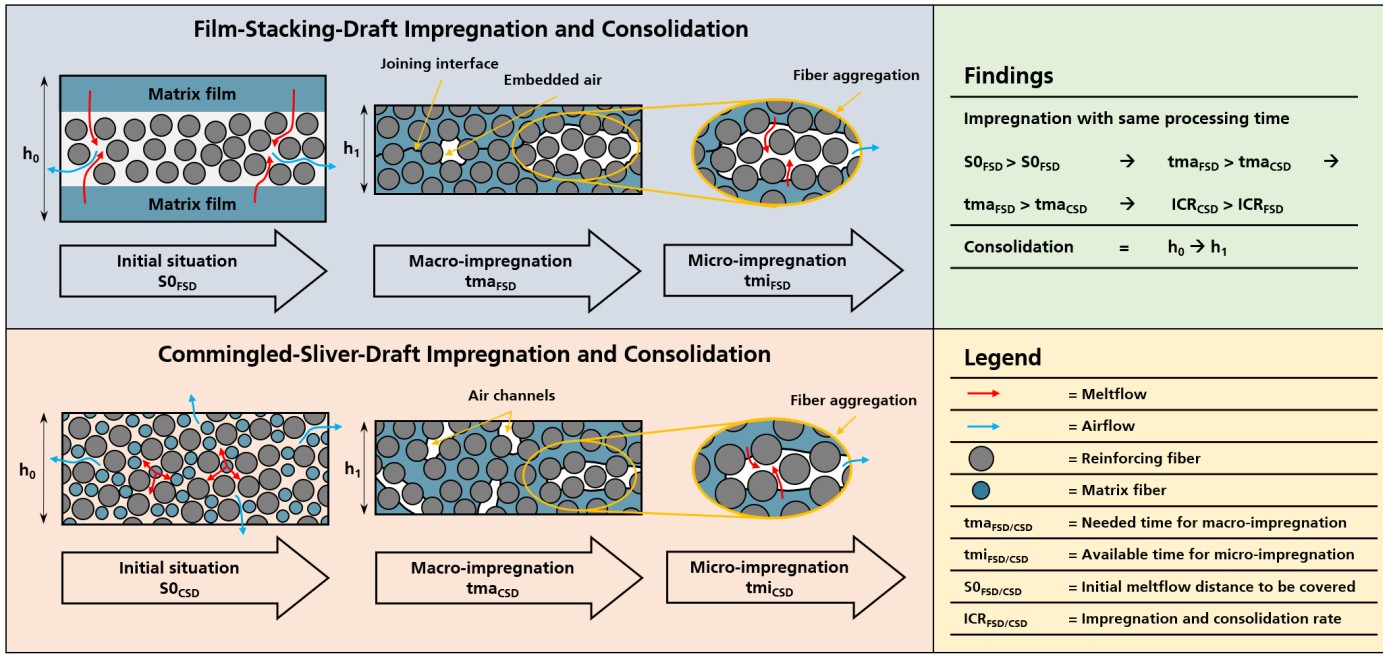

**Figure 9.** Schematic illustration of the suggested initial way of impregnation between film-stacking and commingled sliver composite.

The critical impregnation and consolidation time is the decisive criterion for maximum production speed. In other words, the minimum time required to melt the matrix, impregnate the reinforcing fibers and eliminate voids from the composite can be understood as a measure of the productivity of the process, depending on the desired degree of consolidation for later applications (in this study the aim is a fully consolidated tape). The production speed of the FSD process compared to the CSD process is a very limiting factor [53]. Further studies are needed to evaluate the limits of this production method. In this context, an increase in fiber volume fraction (here 38% by volume) without quality acceptance is also plausible. In addition, the resource efficiency in the CSD process is advantageous due to there being significantly less waste at the edge areas of the CSD tapes compared to the FSD tapes.

### 3.2.2. Mechanical Analysis

The advantageous impregnation is also evident when comparing the flexural properties of CSD laminate with FSD laminate. Hence, NFRP are often used for applications subject to bending loads or impact resistance, and this effect should be emphasized. In Figure 10, the mechanical results of tensile tests according to DIN 527-5 and bending tests according to DIN 14125 are displayed. The result of fiber structure optimization using optical analysis can be confirmed by comparing the bending properties of the FSD laminate test series. Bending properties are strongly influenced by the morphology and impregnation quality of the reinforcement. The best-found processing parameters tendencies (high draft ratio and many input slivers with low density) show the highest bending strength in descending order: FSD.3, FSD.6, FSD.9. Since a high DR was chosen for all three, the LDS can be defined as the most important influencing factor here (depending on NB). The low scattering of the flexural strength, which was also observed for the tensile strength, underlines the advantageous fiber architecture of FSD.3. The curves of the FSD.6 and FSD.9 laminates show strong fluctuations, while the curves of FSD.3, consistent with the best results of the structural optical analysis, are the most uniform curves with the highest flexural stress but similar modulus. Thus, this confirms the effect of a better impregnation and consolidation quality due to optimized permeability of the fibrous preform using the investigated draft parameters.

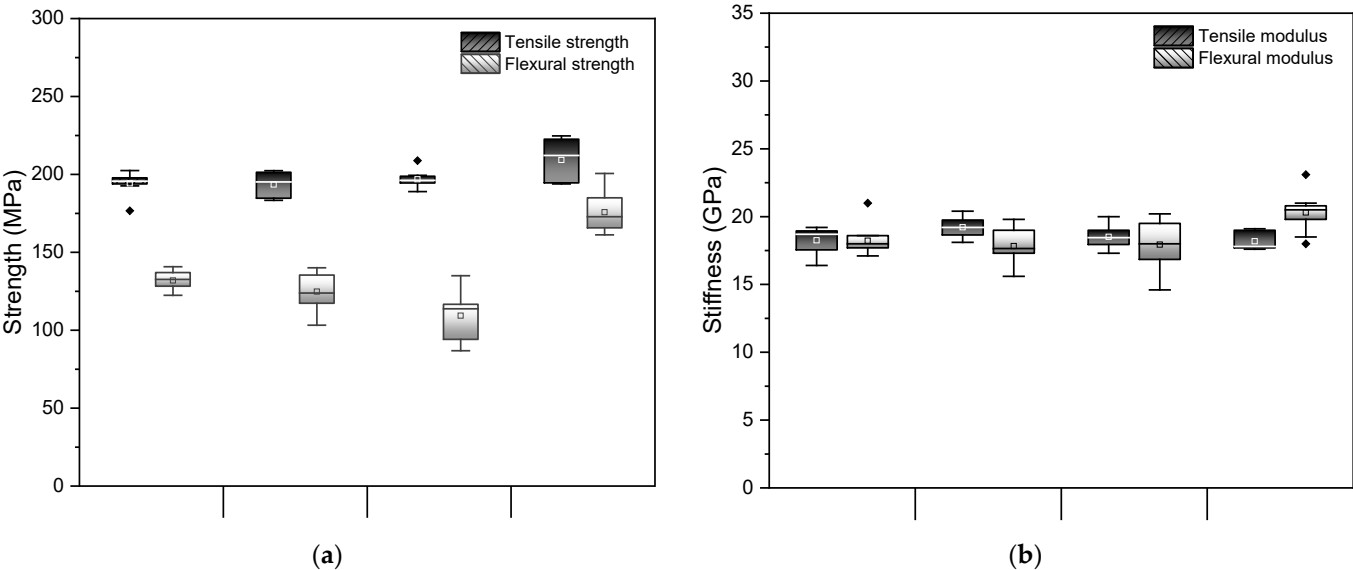

(**a**)                    (**b**)

**Figure 10.** Mechanical results according to tensile tests DIN 527-5 and bending tests DIN 14125: (**a**) tensile and flexural strength of chosen laminates; (**b**) measured tensile and flexural modulus of chosen laminates.

An improvement of 34% in flexural strength and 10% in flexural modulus can be detected for the CSD compared to best performed FSD laminate (FSD.3). Figure 11 displays averaged stress–strain curves with standard deviation as the scattering field of the carried-out bending tests. The tensile properties of composites are mainly dominated by fiber properties—no clear tendencies or conclusions can be derived from the tensile test results. Analogous to the highest flexural properties, the bending curve of CSD laminates shows a desired course with high rise and peak as well as good uniformity and low scattering.

The area under the curves as a measure of energy absorption is slightly higher for FSD.3 (+12%) in this chart than for CSD. However, the impact behavior of CSD is probably advantageous due to the steep slope and needs to be investigated in further studies. Strain could only be interpreted until 8%. Comparable composite materials and their static mechanical properties are listed in Table 7.

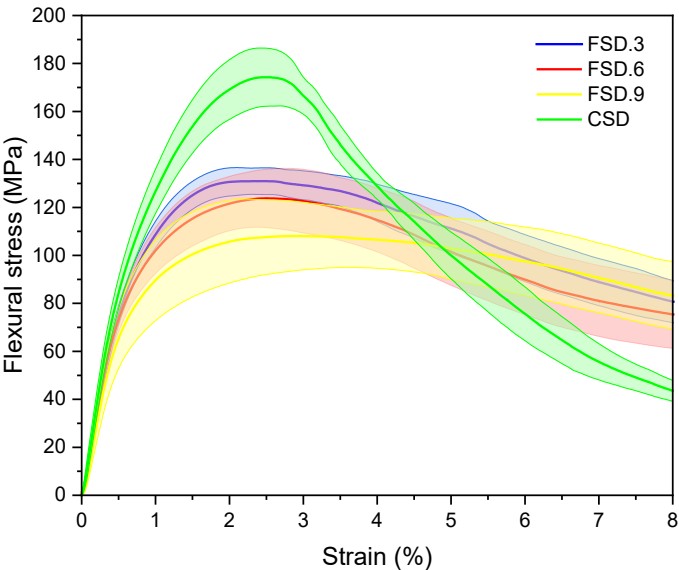

**Figure 11.** Results from bending tests according to DIN 14125: Averaged stress–strain curves with standard deviation as scattering.

**Table 7.** Mechanical properties of similar composite materials with 50 weight-% reinforcing fibers.

| Reference | Composition | Flexural Modulus | Flexural Strength | Tensile Modulus | Tensile Strength |
|---|---|---|---|---|---|
| FSD.3 | Flax/PP | 18.0 GPa | 131 MPa | 19.0 GPa | 194 MPa |
| CSD | Flax/PP | 20.0 GPa | 175 MPa | 19.0 GPa | 200 MPa |
| [40] | Flax/PP | 22.0 GPa | 132 MPa | 26.0 GPa | 125 MPa |
| [54] | Flax/PA12 | - | - | 23.1 GPa | 222 MPa |
| [55] | Flax/PP | - | - | 16.0 GPa | 160 MPa |

The fiber–matrix adhesion was not optimized for any of the materials in Table 7. Although various methods are known and reviewed. For the use of natural fibers in composites, methods of altering the fiber–matrix adhesion are summarized in [8,24–26,30,37,45,56]. Mechanical properties of natural fibers vary greatly due to natural fluctuations, measurement conditions, origin or preparation and processing, among other factors. For an estimation of the reinforcement potential of natural fibers, characteristic values can be compared and taken from [6,8,9,24,29,31,37,57–59]. Because natural fibers are very sensitive to moisture, it can be advisable to dry the most natural fibers before processing them into composites; however, this was not implemented in this work. The production temperature should be kept as high as necessary (to maximize the melt flow) and as low as possible to avoid damaging the natural fiber [39]. In addition to high-temperature exposure, damage can also occur during the mechanical processing of natural fibers inter alia through the use of a drawframe, especially the gillbox type with needle bars. The most well-known problem, in general, is the appearance of kinks along the fiber, but fiber length reductions or fanning effects also occur. Studies about failure modes and defects of NFRP can be found in [58,60]. Different production methods for unidirectional natural fiber-reinforced thermoplastics have been developed with various approaches [31,40,45,53,54,59,61,62], especially in recent years. This proves a renewed and increased interest in harnessing the technology.

## 4. Conclusions

This work has demonstrated the integration and use of a gillbox to provide a fibrous preform from natural fibers and commingled natural and polymer fibers for unidirectional composite tape production. To ensure sufficient porosity of the preform with maximized

free contact surface of the reinforcing fibers, the following draft parameters worked best for defibering the drafted slivers: high draft ratio of $10 \pm 2$, low linear density and high quantities of slivers, which are processed simultaneously. The evaluation of tape structure using gray scale analysis appears to be a well-applicable method for quality assurance. Better micro impregnation with the same process time was proved using premixed reinforcing and matrix fibers due to the reduced flow path of the polymer, superior fiber distribution and more channels to evacuate air and moisture. This results in improved mechanics due to better load transfer. An improvement of 34% in flexural strength and 10% in flexural modulus can be detected for the laminates made from the commingled-sliver-draft process, compared to the structurally optimized film-stacking-draft laminates. The upstream mixing also enables the increase in production speed (depending on the degree of consolidation) and fiber volume fraction up to a certain degree. Since the optimization of the mechanical properties of the natural fiber-reinforced composites was not the focus of this work, the range of properties still offers the potential for improvement, e.g., by applying bonding agents or pretreatments for subsequent studies.

**Author Contributions:** Conceptualization, L.K., K.K. and S.C.; methodology, L.K., K.K., D.H. and S.C.; software, D.H.; validation, A.M. and P.H.; formal analysis, D.H.; investigation, D.H. and S.C.; data curation, D.H.; writing—original draft preparation, D.H.; writing—review and editing, P.H., A.M. and H.C.; visualization, D.H.; supervision, S.C., I.J., A.M. and H.C.; project administration, L.K., K.K., S.C. and I.J.; funding acquisition, L.K., K.K., S.C., P.H. and I.J. All authors have read and agreed to the published version of the manuscript.

**Funding:** This research was funded by the German Federal Ministry of Food and Agriculture (BMEL), grant number 22012417.

**Institutional Review Board Statement:** Not applicable.

**Informed Consent Statement:** Not applicable.

**Data Availability Statement:** All data included in this study are available upon request by contact with the corresponding author.

**Acknowledgments:** Special thanks to SachsenLeinen GmbH for the permanent loan of the gillbox.

**Conflicts of Interest:** The authors declare no conflict of interest.

## Abbreviations

| | | |
|---|---|---|
| DR | Draft ratio (determines output ratio) | Process parameter |
| LDS | The linear density of slivers (material input) | Process parameter |
| NB | Number of slivers (material input) | Process parameter |
| ADP | The areal density of fibrous preform (material output) | Process parameter |
| GO | Gilling operation (material transformation) | Process parameter |
| FSD | Film-stacking-draft | Process type |
| CSD | Commingled-slivers-draft | Process type |
| AGV | Average gray value | Analytic parameter |
| CV | Coefficient of variation | Analytic parameter |
| MD | Material defects | Analytic parameter |
| MA | Material accumulations | Analytic parameter |

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
