# Peer review of "Manufacturing and Analysis of Natural Fiber-Reinforced Thermoplastic Tapes Using a Novel Process Assembly"

_sustainability, doi:10.3390/su15076250_

Round 1

Reviewer 1 Report

According to table 8, could you please explain why (row 2 and row 3) the flexural modulus and tensile modulus are lesser than the sample compared?

According to table 8, could you please explain why (row 2 and row 3) the flexural strength and tensile strength are much higher than the sample compared?

In figure 2, it is "Silvers containing reinforcing and matrix fibers" I think it should be "Silvers containing reinforcing fibers and matrix". Please check whether I am right or wrong. In case I am wrong, please explain what is "matrix fibers". 

How the binding ability of silver with fiber and matrix is achieved? Could you please explain it in detail?

According to table 4, the production parameter, which is the heating temperature, is 190 degrees Celsius. Could you please explain in detail how it will not affect the fiber at this elevated temperature? 

Author Response

Dear Sir or Madam,

thank you for the review. The response is attached. Please have a first look at response 2 to avoid confusion. I hope my answers are understandable and comprehensible.

Kind regards

David Hartung 

Reviewer 2 Report

The manuscript is good and can be accepted for publication upon incorporating the following comments. Hence revision is suggested at this stage, and the comments are as follows.

1.      The introduction should have the natural fiber, its types, and various factors influencing the performance of natural fibers. The papers below will give an idea of such discussions in the introduction of their study.

https://www.tandfonline.com/doi/abs/10.1080/15440478.2021.1875353

https://www.sciencedirect.com/science/article/abs/pii/S0141813018347135

https://www.tandfonline.com/doi/abs/10.1080/15440478.2019.1642826

2.       The previously published literature related to this work has to be reported. It is very important because it can spotlight the research gap, which will give the novelty of this work.

3.      How were the Table 3 & 4 parameters set? Give some information about it.

4.      SEM analysis of the mechanically tested composite may be given.

5.      Conclusions should be more precise.

Author Response

Dear Sir or Madam,

thank you for the review. The response is attached. I hope my answers are understandable and comprehensible.

Kind regards

David Hartung 

Reviewer 3 Report

The manuscript is well written and all the procedures appropriately detailed.   The results will be of importance in the development of natural fiber filled composites such as UD tapes.  I do not have any changes to recommend. 

Author Response

Dear Sir or Madam,

thank you very much for the review. 

Kind regards

David Hartung 

Reviewer 4 Report

The manuscript corresponds to the Sustainability. The Introduction (Lines 14-31) and the list of references (Lines 488-627) are quite complete.

The results are impressive, and the presentation of the work is excellent.

The methodology of the study is described in sufficient detail; however, it can still be improved.

The text is reasonably clear and easy to read.

All structural units of the manuscript are logically interconnected, except the conclusion, which should be more detailed.

The manuscript contains important scientific results for practice. 

 Comments and suggestions:

1.       In general, the SEM Images of the samples must be included.

2.       Fig 8 and fig 10: Τhe photos should be made bigger so that the readers can see the better details

3.       Conclusion must be more detailed.

Author Response

(The authors gave the same response as above.)
